# Background colour preference in juvenile European lobster (*Homarus gammarus*)

Matt E. Bell[1,2,*], Nicholas Kuvaldin[1,2,*], Erik Romero Frontaura[1,2,*], Felix K. A. Kuebutornye[1,3], Paula Costa Domech[1,2], Carlos Espírito-Santo[4], Rodrigo O. A. Ozório[4] and Alex H. L. Wan[1,2,‡]

## ABSTRACT

The European lobster (*Homarus gammarus*) is a commercially valuable species facing increasing anthropogenic pressure. As hatchery-based restocking and aquaculture expand, optimising artificial rearing conditions, particularly environmental factors like habitat colouration are essential for improving welfare. The present study investigated background colour preference in juvenile (poststage V) lobsters. Custom triangle segments were 3D printed with different-coloured filaments and assembled into hexagonal chambers. In sub-trial 1, each chamber contained six colour: black, red, blue, green, white, and yellow. Lobster behavioural responses were measured using three metrics: time spent (duration), number of incidental movements, and final colour settlement. These metrics had five behavioural parameters: mean duration, total duration, colour preference ratio, final colour settlement and incidental movement. An assessment of these metrics and behavioural parameters revealed that lobsters had a similarly significant preference for black and red background colours. To discern whether there is a higher preference for one colour over the other, sub-trial 2 was performed, using equal black and red segments. Sub-trial 2 indicates that the preference for red is marginally greater than black under the same metrics. These findings contribute to a better understanding of lobster environmental preference and can inform the design of aquaculture systems to enhance animal welfare.

KEY WORDS: Lobster, Colour preference, Habitat colouration, Rearing conditions, Animal welfare, Stock enhancement

## INTRODUCTION

The European lobster (*Homarus gammarus*) is a commercially valuable species with a fisheries value exceeding €40 million in 2021 (Hinchcliffe et al., 2022). It is also a species currently under pressure from several environmental and anthropogenic factors that are reducing its overall fitness. Such factors include overfishing, rising sea temperatures, habitat loss, and increasing offshore renewable energy development (Hinchcliffe et al., 2022). With larval survival rates seldom exceeding 20% from the planktonic stages (Addison and Bannister, 1994; Daniels et al., 2010; Ellis et al., 2015; Powell et al., 2017; Goncalves et al., 2023; Bell et al., 2025), addressing environmental stress factors, such as enclosure colour, may reduce juvenile mortality in hatcheries. This is especially the case during vulnerable stages, such as ecdysis, notably the transitional stage from stage II to stage III, when mortality peaks (Addison and Bannister, 1994).

In the natural environment, colour plays an important role in habitat settlement (Kelber and Osorio, 2010). Animals often respond to environmental background colours, using them as cues for mating behaviour or as warning signals, depending on species-specific association (Stevens and Ruxton, 2012). While many studies focus on animal body colouration, it is often the contrast between an animal and its background that drives ecological interactions and behavioural responses. For example, the poison dart frog (*Dendrobates pumilio*) utilises vivid body colours to deter predators by contrasting blue, yellow, or red (Wang and Shaffer, 2008). While in sexual selection, male blue crabs (*Callinectes sapidus*) show a preference for females with red claws over those with orange ones (Baldwin and Johnsen, 2012). The American lobster (*Homarus americanus*) exhibits darker, bluer colouration, which is influenced by environmental conditions (Tlusty et al., 2009). Understanding how colour perception impacts natural selection, mating capability, and survivorship is crucial for lobster autecology and conservation-based restocking initiatives.

Colour in the aquatic environments plays a pivotal role in influencing animal behaviour, survival strategies, navigation of surroundings and forms of communication (Franklin et al., 2020; Twort and Stevens, 2023). Substrate colour, for instance, can significantly affect growth performance in the whiteleg shrimp (*Litopenaeus vannamei*), with individuals reared on red or yellow substrates showing higher specific growth rates and feeding efficiency than those reared on blue or green substrates (Luchiari et al., 2012). Studies on other aquatic species, such as thinlip mullet larvae, have demonstrated that tank colour can modulate physiological and behavioural responses via photoreceptive pathways, thereby affecting stress hormone levels, growth, and aggression (El-Sayed and El-Ghobashy, 2011). The colour of the tank also alters the physical light environment by influencing intensity, reflectance, and absorption (Lesmana et al., 2021). Such changes in lighting can affect lobster welfare, as prolonged exposure to intense illumination may trigger stress responses and increase mortality (Fitch and Lankford, 2013). Mynott et al. (2018 preprint) investigated camouflage in juvenile European lobsters by alternating black and white rearing backgrounds and assessing

[1]Aquaculture and Nutrition Research Unit (ANRU), Carna Research Station, School of Natural Sciences and Ryan Institute, University of Galway, Carna, Co. Galway H91 V8Y1, Ireland. [2]Aquaculture and Nutrition Research Unit (ANRU), School of Natural Sciences and Ryan Institute, Annex building, University of Galway, Galway, Co. Galway H91 TK33, Ireland. [3]University of South Bohemia in České Budějovice, Faculty of Fisheries and Protection of Waters, South Bohemian Research Center of Aquaculture and Biodiversity of Hydrocenoses, Institute of Aquaculture and Protection of Waters, České Budějovice, Czech Republic. [4]CIIMAR/CIMAR-LA, Interdisciplinary Centre of Marine and Environmental Research, University of Porto, Matosinhos 4450-208, Portugal.
*These authors contributed equally to this work.

‡Author for correspondence (alex.wan@universityofgalway.ie)

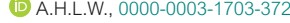 A.H.L.W., 0000-0003-1703-3725

how well they blended in when viewed by a predator species (pollack, *Pollachius pollachius*). This study suggests that darker backgrounds not only reduce visual contrast and improve camouflage but may also align with lobsters' inherent photophobic behaviour, an active preference for low-light environments (Lesmana et al., 2021). Together, these findings highlight both the proximate mechanism (light avoidance) and the ultimate function (camouflage and predator evasion) underlying the preference for darker substrates.

Beyond its broader ecological significance, colour also shapes specific behavioural processes such as habitat selection, foraging, and predator avoidance (Kelber and Osorio, 2010). While crustacean research has largely focused on how environmental variables shape carapace pigmentation (Yeap et al., 2022), much less attention has been paid to how these animals perceive and react to background colour in their environments. The perception of colour varies significantly among aquatic species, shaped by visual adaptations, behaviour patterns (e.g., diurnal/nocturnal activity), structural habitat complexity, and prevailing light conditions (de Busserolles et al., 2020; Caves and Johnsen, 2021). Given these factors, further investigation of environmental colour and visual perception in larval lobster rearing, particularly in aquaculture, is warranted. This study aimed to determine the most suitable environmental colour (red, green, blue, yellow, black, and white) for juvenile European lobsters (post-stage V) by recording their movements across colour chambers. This approach could enhance physiological plasticity and support juvenile lobsters' acclimation prior to release, ultimately improving restocking outcomes. Identifying preferred colours may inform the design of lobster aquaculture systems by enabling environmental enrichment that mimics natural habitats while maintaining appropriate animal husbandry practices, e.g., allowing the hatchery operator to see the animals still.

## RESULTS
### Sub-trial 1
In sub-trial 1, juvenile European lobsters showed measurable preferences across the six colour segments. Lobsters spent the longest mean duration in the red chamber and the black chamber ($P<0.05$, Fig. 1A). The lowest mean durations were recorded for blue, green, yellow and white. Total duration spent in each chamber showed a similar trend, with black receiving the highest cumulative time (6640 min), followed by red (5600 min). Statistical comparison was not performed for this metric because the values

represent pooled durations across all individuals (Fig. 1B). The highest movement incidence (Fig. 1C) was observed in the black chamber, with a preference ratio of 0.226, followed closely by the red chamber (0.217). Both black and red were entered significantly more often than the other colours tested ($P<0.05$). Final settlement preference revealed similar findings. A chi-square test confirmed that the frequency of final settlement in black and red chambers was significantly different from random expectation ($P<0.0001$), reinforcing the behavioural result from the duration metrics. The final settlement preference is included in Figs S1 and S2.

### Sub-trial 2
To further examine whether red or black was more preferred in direct comparison, sub-trial 2 used chambers with equal red and black segments. The pattern of mean duration showed a slightly higher mean for red compared to black (Fig. 2A). However, paired *t*-tests revealed no significant difference in mean duration between the two colours ($P>0.05$). Similarly, no significant difference was observed for final settlement frequency in sub-trial 2. Colour preference ratios were shown to be significantly higher for red than for black ($P<0.05$, Fig. 2C). As with sub-trial 1, no statistical testing was performed on total duration metrics in sub-trial 2, since these data represent cumulative observations across the groups (Fig. 2B).

### CIELAB colour space correlation
To quantify the physical characteristics of each colour segment, the CIELAB colour space was used to derive values for L* (lightness), a* (red-green axis), and b* (yellow-blue axis). Regression analyses were performed to correlate these colorimetric properties with lobster behavioural metrics (final settlement, mean duration, and incidental movement). Scatter plot analyses between colour preferences and L* (Fig. 3), a* (Fig. 4), and b* (Fig. 5) values showed weak associations: $R^2$ values were 0.358 for L*, 0.069 for a* and 0.203 for b*, indicating no strong predictive relationship between these colour metrics and lobster preference.

## DISCUSSION
The results of this study revealed a clear and consistent behavioural preference in juvenile European lobsters for black and red environments over the other colours tested. While both colours were strongly favoured in terms of time spent and movement, red exhibited a small but statistically significant advantage over black when analysed through the colour preference ratio. This difference indicates that red may elicit an additional behavioural cue beyond

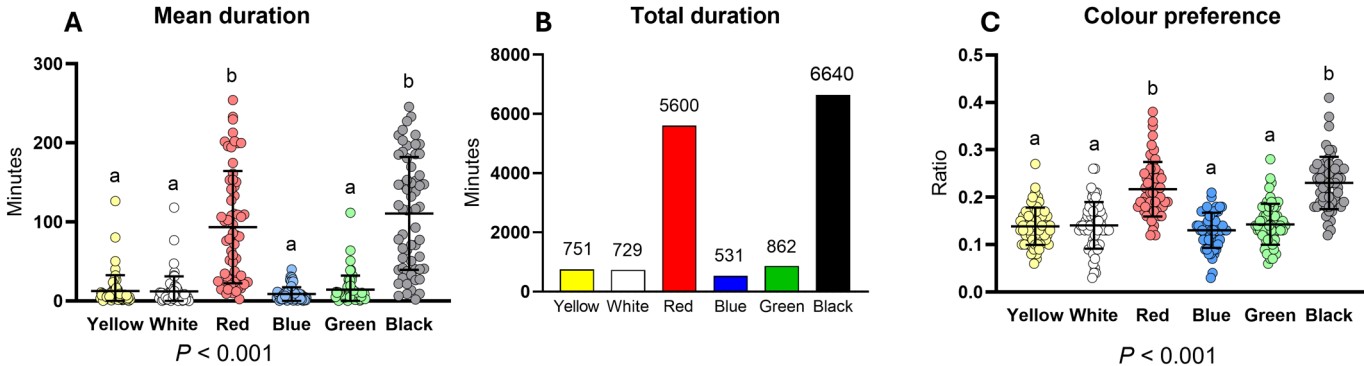

**Fig. 1. Colour preferences measured via various design parameters in sub-trial 1.** (A-C) Mean (A) and total (B) duration in each chamber, and colour preference ratio (C). In mean duration and colour preference, long horizontal bars represent the mean, and short horizontal bars represent ±s.d. Different letters indicate significant differences among experimental groups (Friedman test, followed by Dunn's multiple-comparisons test, $P<0.05$); $N=60$. No statistical comparison was conducted for total duration, as these values represent cumulative data across all individuals.

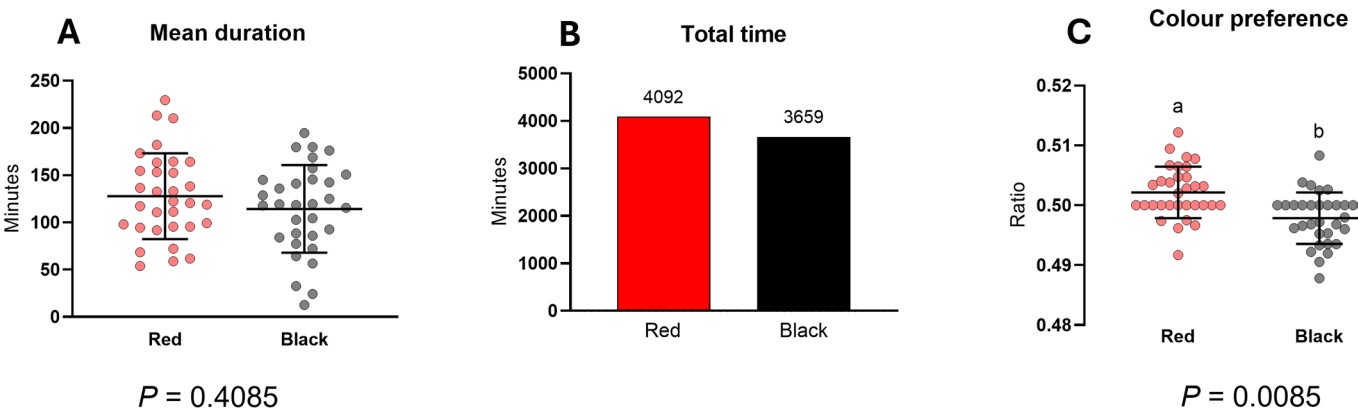

**Fig. 2. Colour preferences measured via various design parameters in sub-trial 2.** (A-C) Mean (A) and total (B) duration in each chamber, and colour preference ratio (C). In mean duration and colour preference over a 4 h period for each lobster experiment, long horizontal bars represent the mean, and short horizontal bars represent ±s.d. Different letters indicate significant differences among experimental groups (paired $t$-test, $P<0.05$); $N=30$. No statistical comparison was conducted for total duration, as these values represent cumulative data across all individuals.

low luminance alone. Thus, the primary interpretation is that low-light environments are preferred overall, with red potentially offering an enhanced cue under the specific experimental conditions. This study is the first to investigate colour preference in European lobster, a species of significant ecological and commercial importance that is currently subject to overfishing (Macneil et al., 1997; van der Meeren, 2005). Our findings contribute to the development of colour-based enrichment strategies for hatchery design, aiming to improve larval survival and welfare in captive settings (Agnalt et al., 1999). Although we did not test whether the preference is innate or shaped through experience, the lobsters used were reared entirely in uniform background conditions with no prior exposure to coloured substrates. This supports the interpretation that the observed preferences are unlikely to reflect learned behaviour but rather may be innate or developmentally constrained. Future studies could help clarify this distinction through controlled rearing environments with varied chromatic exposure.

**Visual ecology and colour preference in European lobsters**
Research on crustacean visual pigments shows that many decapods have very limited sensitivity to long-wavelength red light. In the American lobster (*H. americanus*), visual pigment rhodopsin peaks at ~515 nm and converts to metarhodopsin at ~490 nm, indicating poor detection of red wavelengths (Bruno et al., 1977; Wald and Hubbard, 1957). Similarly, Norway lobster (*Nephrops norvegicus*)

exhibits rapid photoreceptor degeneration under moderate red-light exposure, highlighting a narrow tolerance to long wavelengths and potential vulnerability to artificial lighting (Loew, 1976). In aquaculture settings, this implies that the standard lighting used could compromise visual function, orientation, and behaviour in Norway lobster, raising concerns about the welfare and performance of this species in captivity.

In contrast to *Nephrops*, *Homarus* spp. demonstrate greater resilience to light exposure due to their capacity for photoregeneration and recovery of visual pigments (Bruno et al., 1977). In *Homarus*, dark-adapted eyes contain predominantly rhodopsin, and, following exposure to actinic light, metarhodopsin is regenerated in darkness with a half-life of 25-55 min, depending on water temperature (Bruno et al., 1977). This suggests a temperature-sensitive recovery mechanism that may buffer *Homarus* against occasional high-intensity light environments, unlike in *Nephrops*. Together, these studies reinforce that the visual ecology of *Homarus* enables functional sensitivity to dim, green-blue wavelengths, while their behavioural preferences for red and black may be driven by non-chromatic cues such as luminance, contrast, or perceived refuge value. Thus, red may serve more as a proxy for shelter than a visually distinct stimulus, which aligns with the known physiology of lobster rhabdomeric photoreceptors and their natural crevice-dwelling behaviour.

Although European lobsters possess peak visual sensitivity in the green spectrum (~520 nm) and reduced sensitivity to long-wavelength

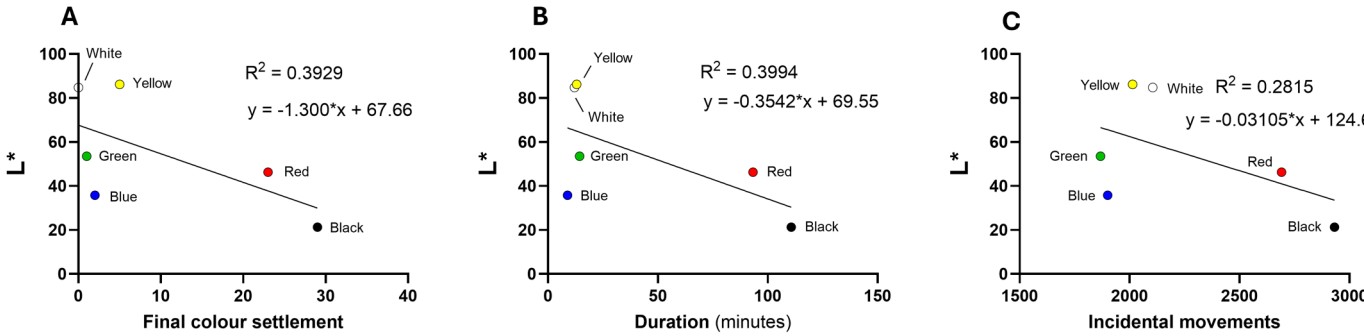

**Fig. 3. Scatter plots on the relationship between L\* [(+)whiteness/(−)blackness] and behavioural responses of European lobsters.** (A-C) Final colour settlement (A), duration in each colour chamber (B), and number of incidental movements into each chamber (C). Regression lines and $R^2$ values indicate the strength of linear correlation; data represent means from $N=60$ individuals.

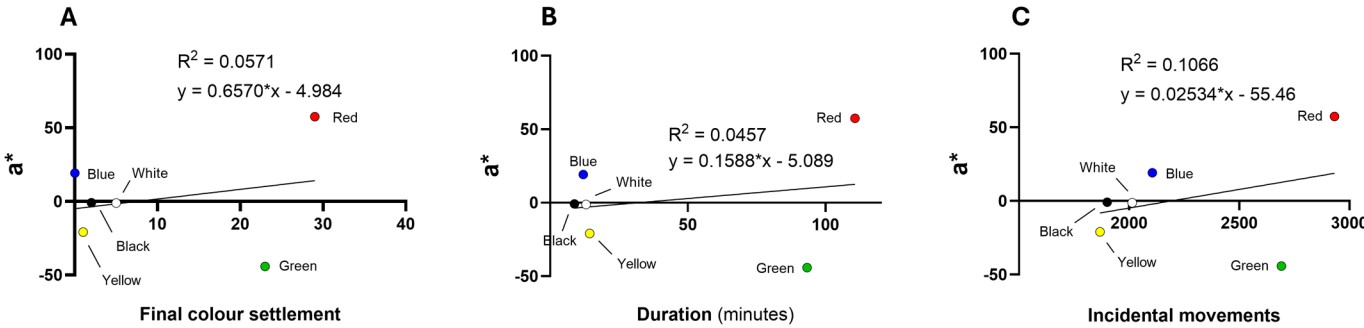

**Fig. 4. Scatter plots on the relationship between a\* [(+)redness/(−)greenness] and behavioural responses of European lobsters.** (A-C) Final colour settlement (A), duration in each colour chamber (B), and number of incidental movements into each chamber (C). Regression lines and $R^2$ values indicate the strength of linear correlation; data represent means from $N$=60 individuals.

red light (>600 nm) (Kennedy and Bruno, 1961), the observed preference for red over blue in this study seems counterintuitive based purely on photoreceptor tuning. This paradox may indicate that lobsters are responding not primarily to chromatic hue, but to non-chromatic features associated with red substrates, such as low luminance, high contrast, or association with sheltered crevices. In other words, red may act as a proxy for dark refuge, rather than a true spectral cue. This interpretation aligns with findings of poor red-light perception in *Homarus* spp. (Bruno et al., 1977) and behavioural sensitivity to luminance contrast rather than hue in other crustaceans (Franklin et al., 2020). The stronger behavioural avoidance of blue, which is more abundant in open water, further supports the hypothesis that refuge-seeking underlies substrate choice in benthic crustaceans such as lobsters. Interestingly, red is considered a weak and ineffective visual stimulus for aquatic species (Weiss et al., 2006) because it is rapidly absorbed by water. The current study observed a clear preference for red and black over green. This is surprising, given the rarity of red light in natural underwater environments due to its rapid absorption in water. Nevertheless, it also showed a relatively higher preference for red and black in sub-trial 1 than for other colours, such as blue, which is more abundant in marine environments due to the scattering of shorter wavelengths (Stramski et al., 2004). Preference ratios also demonstrated minimal variation between black and red, a difference of 4.06%. Furthermore, incidental movement was marginally higher for black than for red, with a 0.62% difference, further evidencing the close behavioural response to these colours. This unexpected preference for red may be inherently correlated with the red pigment astaxanthin present in crustaceans (Krawczyk and Britton, 2001). However, further investigation is necessary to determine whether this behavioural response has a biochemical basis.

The preference for red backgrounds in European lobsters may reflect an innate, phylogenetically conserved trait, possibly associated with camouflage or environmental familiarity during early developmental stages. Although red wavelengths have been suggested to evoke restorative responses in other species, particularly in humans (Borges et al., 2014; Wahl et al., 2019), there is currently no direct evidence that similar neurological pathways exist in crustaceans. As such, any comparison of red-blue light effects across taxa must be interpreted with caution, i.e. lobsters have poor sensitivity to red wavelengths and may therefore perceive red backgrounds as darker than blue ones. In European lobsters, the observed preference for red may be more plausibly explained by non-chromatic visual factors, such as low luminance and high contrast, which enhance their ability to locate shelter, consistent with their benthic, crevice-seeking ecology. While such mechanisms have not been confirmed in lobsters, these associations may relate to the nocturnal and crevice-dwelling behaviour of European lobsters (Spanier and Zimmer-Faust, 1988), wherein red or darker environments offer reduced visibility to predators and thus signal a lower-risk, refuge-like setting.

It is important to note that this study assessed acute colour preference, with each animal tested only once in a single behavioural trial. While this approach provides insight into immediate substrate choice, it does not capture long-term settlement or habituation under continuous exposure. Therefore, although our results suggest black and red are preferred in short-term exploratory contexts, further research is needed to determine whether these preferences persist over extended periods and whether they translate into differences in growth, stress physiology, or survival in aquaculture settings.

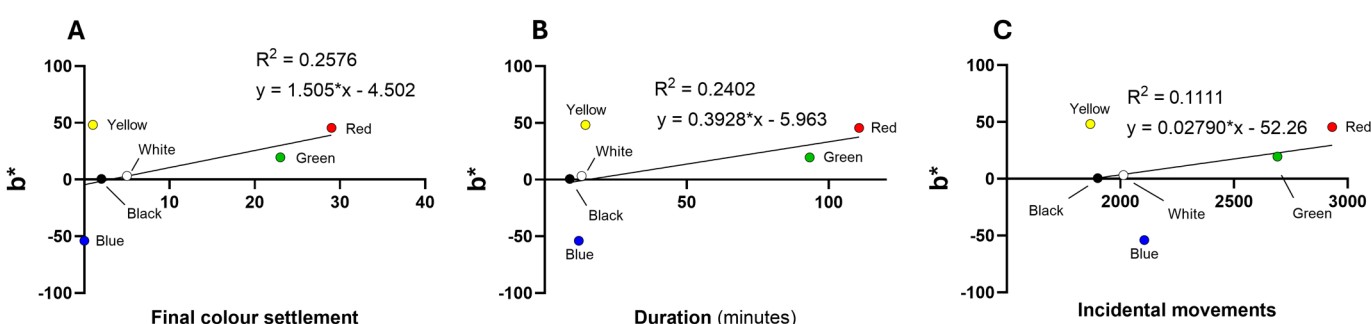

**Fig. 5. Scatter plots on the relationship between b\* [(+)yellowness/(−)blueness] and behavioural responses of European lobsters.** (A-C) Final colour settlement (A), duration in each colour chamber (B), and number of incidental movements into each chamber (C). Regression lines and $R^2$ values indicate the strength of linear correlation; data represent means from $N$=60 individuals.

The present findings indicate that colour preference in European lobsters is species-specific and likely linked to their benthic ecology. While environmental colour has been shown to influence behaviour, growth, and stress responses in other aquatic species, including rainbow trout (*Oncorhynchus mykiss*) and Nile tilapia (*Oreochromis niloticus*), the direction of these responses varies across taxa (Luchiari and Pirhonen, 2008; Colson et al., 2021; Maia and Volpato, 2013). In contrast to many fish studies, juvenile European lobsters in the present study consistently favoured red and black backgrounds, suggesting that preference is shaped by ecological context rather than universal colour effects. This supports the need for species-specific evaluation of environmental colour in aquaculture settings. For European lobsters, this discrepancy is most likely a function of their benthic ecology and crevice-dwelling habits, where darker substrates provide camouflage and reduced predation risk (Spanier and Zimmer-Faust, 1988), while blue and green colours, which are more abundant in pelagic environments (Stramski et al., 2004), may signal exposure rather than refuge. Together, these results emphasise the importance of designing aquaculture systems that match the specific behavioural and ecological requirements of the focal species, rather than extrapolating from other taxa.

## Chromaticity, behavioural metrics and aquaculture implications

To characterise the visual properties of the coloured chambers in the present study, CIELAB analysis was employed, revealing weak correlations between these values and lobster behavioural variables. Of the three axes, L* (lightness) showed the strongest link to behaviour, indicating that responses may be more strongly influenced by perceived brightness (or shade) than hue per se. Although both black and red chambers exhibited low luminance levels, red was selected significantly more often than black in the preference ratio analysis, suggesting that a cue beyond darkness alone may be driving this preference. A related complication is that, under the study's lighting, the blue chamber had the lowest measured luminance yet was not preferred. This discrepancy likely reflects differences between instrumental luminance and lobster-perceived brightness, i.e. lobsters are more sensitive to blue-green wavelengths (∼480-520 nm) and have very poor sensitivity to long-wavelength red light. As a result, a blue surface that is objectively darker may be perceived as relatively brighter, whereas red, despite having a higher L* value, may be perceived as the darkest or most concealing option. This aligns with findings in scalloped spiny lobsters, where individuals showed greater movement towards black and red environments under low-light conditions (Lesmana et al., 2021). It also underscores the importance of considering both luminance and chromatic factors when designing aquaculture environments, as substrate colour may mediate behavioural responses through perceived shade or ecological associations, not just spectral wavelength. Future work should seek to disentangle the interaction between light intensity and colour. Moss et al. (1999), for instance, found that irradiance was a stronger determinant of behaviour than background colour in phyllosoma larvae of rock lobster (*Jasus edwardsii*), with high-intensity light negatively impacting growth and survival. Our use of relatively high irradiance (∼50.5 µmol s$^{-1}$ m$^{-2}$) may have similarly influenced lobster behaviour, highlighting the need to test chromatic preferences under varied light conditions. In addition, longer-term studies (e.g. larvae to pre-release juveniles) should prioritise assessing whether such preferences translate into measurable outcomes in growth, stress physiology, survival, and overall welfare in aquaculture

settings. These investigations would help determine whether colour-based enrichment strategies have practical significance beyond short-term behavioural responses. While beyond the immediate behaviour, colour preference in crustaceans likely reflects integrated sensory processes. Lobsters, for instance, are known to associate colour cues with learned outcomes (Tomina and Takahata, 2012); furthermore, wider decapod species integrate visual signals with tactile and olfactory information during substrate exploration (Kawamura et al., 2016, 2017, 2020). Given the known effects of stress and high-density rearing on lobster immunity and growth (Fotedar and Evans, 2011), the use of behaviourally preferred colours such as red or black may offer a non-invasive strategy to enhance welfare in hatchery systems for stock enhancement, aquaculture on-growing facilities, and holding facilities before market.

## Conclusion

The methodology and environmental considerations required to rear European lobsters for food and stock enhancement remain an open question in terms of production efficiency. To our knowledge, the present study is the first to identify colour preferences in juvenile European lobster, providing substantial insight into how rearing background colour may affect refuge choice and welfare. In summary, juvenile European lobsters showed a marked preference for black and red backgrounds, which likely simulate low-luminance, shelter-like conditions typical of their natural benthic habitats. Although lobsters are visually insensitive to red light, the observed preference suggests they may respond to non-hue-based shelter cues, such as contrast or shadow, rather than chromatic saturation alone, indicating that colour preference in this species may operate through a luminance-driven, refuge-seeking mechanism. Results from both sub-trials showed a consistent preference for red and black over other tested colours. The follow-on trial, sub-trial 2, showed a similar preference for black and red. However, statistical testing indicated that red was most preferred, as indicated by the colour preference ratio. These findings support the hypothesis that darker backgrounds may be perceived as safer and more suitable for lobster larvae settlement, aligning with their benthic, crevice-dwelling nature. Future commercial hatchery designs could benefit from integrating colour-specific modifications to promote not only survival and growth but also broader improvements in animal welfare and physiological status. Therefore, the present study recommends rearing lobster larvae in black, red or a mixture of both colours. However, further research is required to validate long-term survival, growth, and behavioural acclimation under different background colours. Follow-on studies should correlate the effect of background colour with other biological indicators, including feeding behaviour and physiological metrics, and systematically examine the interaction of light intensity and background hue, as well as ontogenetic changes in visual and behavioural responses, to further clarify whether black or red background colour can enhance animal welfare, especially in prolonged holding periods such as those experienced by wholesalers before market and aquaculture production.

## MATERIALS AND METHODS
### Lobster rearing

Juvenile lobsters were obtained from four berried female European lobsters, sourced from a local fisherman caught off the Connemara coastline, Co. Galway, Ireland. Stage I lobster zoea were reared in 70 l hopper systems filled with pseudo-green seawater. This rearing system contained a mixture of microalgae of *Nannochloropsis oculata*, *Dunaliella salina*, and *Isochrysis galbana*. The lobster larvae were initially fed enriched artemia

daily. Upon reaching stage III, the larvae were weaned to formulated particulate feeds (C1/C2, Pacific Trading Aquaculture Ltd., Dublin, Ireland), fed twice daily, and settled individually in separate holding compartments. Once the larvae had developed to post-stage V, i.e. had successfully settled into the benthic juvenile stage, individuals were randomly selected for the colour preference trial.

### Design and production of colour chambers

To assess the lobster colour preference, colour chamber segments were bespoke-designed using Tinkercad software (Autodesk, San Francisco, CA, USA) and then printed using different colour PLA+ (polylactic acid, red, blue, black, green, yellow and white) filaments on an UltiMaker S5 3D printer (Geldermalsen, The Netherlands) at the Make a Space Unit, James Hardiman Library, University of Galway, Ireland. Each set of six printed segments was assembled into a hexagonal chamber (width: 115.4 mm, height: 50 mm, real available volume of one hexagon: 33.98 cm³) and subsequently used in the colour preference trials (Fig. 6). To avoid positional bias, the four colour chambers (black, red, green, and blue) were arranged in a fully randomised order for each trial. This randomisation ensured that each colour was presented in every possible position across the experiment, preventing consistent spatial associations with any single colour.

### Experimental setup

Prior to the colour preference trials, natural seawater (35.40 ppt, measured with a conductivity meter, Seiki Electromechanical Equipment Ltd., Hung Hom, Hong Kong) sourced from the University of Galway's Carna Research Station was filtered through a 10 nm particulate filter and an ultraviolet light steriliser to remove debris and microalgae potentially affecting colouration in the water. Temperature was maintained at 21.24±1.01°C. The treated seawater was used to fill a shallow tray tank (52×42 cm) to a height of 15 cm, and the hexagonal chambers were placed in the tray, ~0.5 cm above the water level, to prevent water disturbance and escapees. Passive water exchange between the chamber and the tray water was achieved through perforated holes at the bottom of each colour segment (Fig. 6). Illumination was provided above each chamber using fluorescent lighting: intensity lux was 2731.37±279.25 (Hobo MX temp/light, ONSET, Bourne, MA, USA), and photon flux density was 48.74±2.95 μmol m² s⁻¹ (380-750 nm, LI180 spectrometer, LI-COR Biosciences UK, Cambridge, UK). To avoid water disturbance to the lobsters, gentle aeration was supplied into the tanks and was placed away from the hexagonal chamber. The same white light source illuminated all chambers; colour differences reflect only substrate colour, not coloured lighting. This study complies with the legislation on the protection of animals used for scientific purposes (Directive 2010/63/EU).

### Colour preference trial and behaviour response analysis

A total of 90 juvenile lobsters were used in sub-trial 1, 0.123±0.023 g lobster⁻¹, 9.272±0.722 mm carapace length (CL) (n=60), and in sub-trial 2, 0.129±0.018 g lobster⁻¹, 9.479±0.557 mm CL (n=30), with a global mean of 0.125±0.022 g lobster⁻¹, 9.350±0.669 mm CL (t-test P=0.235). Prior to the start of each replicate experiment within each sub-

trial, a plastic, semi-transparent cylindrical tube was used to deploy and position the individual lobster in the centre of the chamber unit (Fig. 6). The white tube remained in place for 4 min to allow acclimation before the juvenile lobster was released into the chamber, and the trial commenced and lasted for 4 h. In sub-trial 1, the hexagonal chamber was assembled using one of each of the six different colours. A total of 60 independent behavioural tests were conducted, each using a new lobster (n=60). Given the preferential colour choice observed in sub-trial 1, i.e. red and black, sub-trial 2 was performed using equal proportions of chamber segments of black (i.e. three segments) and red (i.e. three segments). The sub-trial 2 behavioural response test was performed 30 times (n=30), each on a new lobster.

Using the same experimental protocol as sub-trial 1, each behavioural test lasted 4 h as a means of comparison between the two sub-trials. In both sub-trials, each behavioural test was conducted using a new juvenile lobster, and the colour sequence of the chamber segments was assigned randomly using a random number generator to avoid bias and inter-colour effects on the lobster behaviour. The movement of the lobster during the behavioural response test was recorded using a video camera (Connect, Logitech Connect, Lausanne, Switzerland) connected to a computer. Colour preference was noted when more than 50% of the lobster's body was inside the colour boundary of the chamber. For each lobster, the corresponding colour entries and movement durations were tracked and recorded. The durations, incidental movements, and final colour settlement in colour chambers over the 4 h were recorded for later extrapolation. Five key behavioural metrics were quantified: (1) mean duration, the average time spent per entry into each chamber; (2) total duration, the total time spent in each colour chamber per individual; (3) colour preference ratio, calculated by dividing the mean duration per entry for each colour by the total mean duration across all colours for that individual; (4) final colour settlement, the last colour chamber occupied at the end of the trial; (5) incidental movement, the number of times a lobster entered each colour chamber. These values were calculated for each colour chamber and individual lobster and then compiled into a comprehensive dataset for further analysis.

### Morphometric analysis and colourimetry

Immediately after each trial, each lobster was imaged on a grid background using a D80 digital camera (Nikon, Shinagawa City, Tokyo, Japan) with a 105 mm macro lens (Sigma, Kawasaki, Kanagawa, Japan) at 1/80 s, f/5, and ISO 400. The lobster images were measured for the CL (tip of the rostrum to the posterior edge of the carapace) using ImageJ software v1.54h (Schneider et al., 2012). In addition, each of the lobsters was also weighed using a precision balance (0.001 g readability). Colorimetric measurements of the different colour chambers were performed using a Konica Minolta Chroma Meter CR-400 (Konica Minolta Sensing Europe, Warrington, UK), and values were expressed through the CIELAB colour space model (International Commission on Illumination): L* (lightness), ranging from 0 (black) to 100 (white); a* [(+)redness/(−)greenness], and b* [(+)yellowness/(−)blueness] (Plataniotis and Venetsanopoulos, 2000). Five random points were measured on each of the six colour chamber segments to give a mean value.

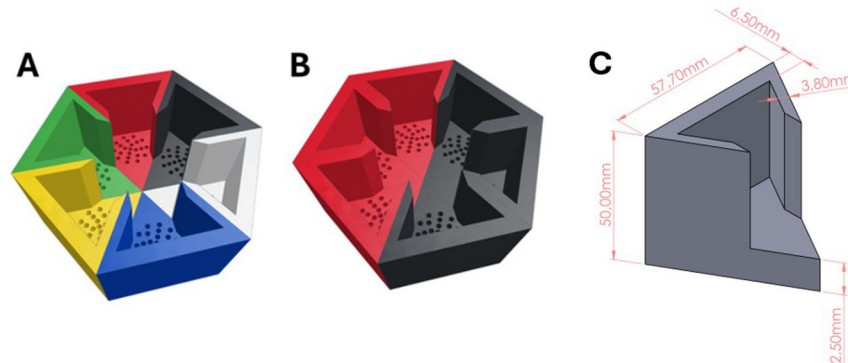

**Fig. 6. Schematic of the 3D-printed colour chambers used in the colour behavioural trial. Each segment is printed individually to allow different colour permutation arrangements.** (A) Sub-trial 1 comprised six different colours: red, blue, black, green, yellow, and white. (B) Sub-trial 2 comprised red and black sub-chambers. (C) The dimensions of the colour chamber segment were produced from 3D printing. Note: A and B have 2 mm perforated holes at the bottom to allow water exchange between the tray and the chamber.

## Statistical analysis

The results were expressed as mean±s.d. Data were analysed using statistical tests that depended on the trial design and the data distribution. For sub-trial 1, normality was assessed using the D'Agostino-Pearson test, and homogeneity of variance was verified using the Levene's test. As the data did not meet parametric assumptions, a nonparametric Friedman test followed by Dunn's multiple-comparisons test was used to determine significant differences in colour preference metrics. Chi-square tests were conducted to assess statistical differences in final colour settlement frequencies. For sub-trial 2, paired data were also tested for normality and homogeneity of variance. A paired $t$-test was applied to compare black and red for all colour preference metrics described above. The level of significance used was $P<0.05$. Pearson's correlation analysis was performed to assess the relationship between the individual L*, a*, and b* colour space values of each chamber (representing the background colour) and the recorded colour preference metrics. All statistical analyses were performed using GraphPad Prism 10 (GraphPad Software Inc., San Diego, CA, USA).

### Acknowledgements

The authors would like to acknowledge the technical and animal husbandry support from Mr Stephen McCusker at Carna Research Station, Ryan Institute, University of Galway. The views expressed do not necessarily reflect those of the European Commission or Special EU Programmes Body (SEUPB).

### Competing interests

The authors declare no competing or financial interests.

### Author contributions

Conceptualization: E.R.F., A.H.L.W.; Data curation: M.E.B., N.K., E.R.F., F.K.A.K., P.C.D., A.H.L.W.; Formal analysis: M.E.B., N.K., E.R.F., P.C.D., A.H.L.W.; Funding acquisition: P.C.D., A.H.L.W.; Investigation: M.E.B., N.K., E.R.F., F.K.A.K., P.C.D., A.H.L.W.; Methodology: E.R.F., P.C.D., A.H.L.W.; Project administration: A.H.L.W.; Resources: A.H.L.W.; Software: A.H.L.W.; Supervision: A.H.L.W.; Validation: C.E.-S., R.O.A.O., A.H.L.W.; Visualization: M.E.B., P.C.D., C.E.-S., R.O.A.O., A.H.L.W.; Writing – original draft: M.E.B., N.K., E.R.F., P.C.D., C.E.-S., R.O.A.O., A.H.L.W.; Writing – review & editing: M.E.B., N.K., E.R.F., C.E.-S., R.O.A.O., A.H.L.W.

### Funding

This study was conducted within the GLIOMACH Project, which was funded by the Sustainable Development Goal Seed Fund 2023, College of Science and Engineering, University of Galway. This work was also supported by the TRACE Project, which is co-funded by the European Union's European Regional Development Fund (ERDF) through the PEACEPLUS Programme, managed by the Special EU Programmes Body (SEUPB). Open Access funding provided by University of Galway under the IReL publishing agreement. Deposited in PMC for immediate release.

### Data and resource availability

All relevant data and details of resources can be found within the article and its supplementary information. The data supporting the findings of this study are available upon request from A.H.L.W.

### Peer review history

The peer review history is available online at https://journals.biologists.com/bio/lookup/doi/10.1242/bio.062203.reviewer-comments.pdf

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
