## [Peer Review File · Biology Open]

Background colour preference in juvenile European lobster (*Homarus gammarus*)

Matt Bell, Nicholas Kuvaldin, Erik Romero Frontaura, Felix K.A. Kuebutornye, Paula Domech, Carlos Espírito-Santo, Rodrigo O.A. Ozório, Alex Wan

DOI: 10.1242/bio.062203

Editor: Lewis Halsey

Review timeline

Original submission:	12 August 2025
Editorial decision:	19 August 2025
First revision received:	21 January 2026
Editorial decision:	30 January 2026
Second revision received:	11 March 2026
Accepted:	13 March 2026

Original submission

First decision letter

MS ID#: bio.062203

MS Title: Background colour preference in juvenile European lobster (*Homarus gammarus*)

Authors: Matt Bell, Nicholas Kuvaldin, Erik Romero Frontaura, Felix K.A. Kuebutornye, Paula Domech, Carlos Espírito-Santo, Rodrigo O.A. Ozório, Alex Wan

I have now reached a decision on the above manuscript.

The reviewer reports are shown at the bottom of this email or can be accessed, together with a copy of this decision letter, by going to:

As you will see, the reviewers raised a number of substantial criticisms that prevent me from accepting the paper at this stage. In particular, they both have concerns about lack of clarity and accuracy in the messaging within the Results and Discussion section.

They suggest, however, that a revised version might prove acceptable, if you can address their concerns. If you think that you can deal satisfactorily with the criticisms on revision, I would be pleased to see a revised manuscript. We would then return it to the reviewers. At this stage, we also ask you to ensure your manuscript complies with our formatting guidelines. Provided you are able to fully address the referees' comments, we are positive about publication of your paper (we accept over 95% of revision submissions) and therefore hope you won't mind any extra work involved in reformatting your manuscript at this point.

Please ensure that you clearly highlight all changes made in the revised manuscript. Please avoid using 'Tracked changes' in Word files as these are lost in PDF conversion.

I should be grateful if you would also provide a point-by-point response detailing how you have dealt with the points raised by the reviewers in the 'Response to Reviewers' box. Please attend to

all of the reviewers' comments. If you do not agree with any of their criticisms or suggestions please explain clearly why this is so.

Reviewer 1

Comments for the author

This manuscript describes useful, clear experiments to test the preference of substrate colour in juvenile European lobster. The experiments themselves appear robust and the results are clear. However, I have major comments on the presentation of the results, their interpretation, and their discussion.

Unfortunately, I don't think the combined results and discussion section works in this instance. Much of the discussion occurs before the reader has been shown the relevant results, and then the same discussion points are re-hashed and repeated later. I find this section far too verbose and repetitive, with many paragraphs that are long and cumbersome. I recommend re-writing the results section clearly and completely, and then moving onto the discussion with clear point-by-point interpretations and recommendations. There is no need to make the same suggestion multiple times. Similarly, there is no need for whole paragraphs that appear to provide a review of the literature without explaining how the mentioned studies relate to the present study.

In terms of interpreting the results, I have two major points of concern: The first of which the text circles around but I don't think ever truly lands on. There is a clear preference for black and red segments, which the authors state is because they represent dark refuges that mimic the "crevice-seeking" behaviour of lobsters. The preference for black segments seems intuitive in that an animal seeks shelter, especially given the stress of handling. However, their own measurements show that the blue segments are darker (or more black) than the red but the text does not offer any explanation for why red is significantly preferred over blue. Is it possible that the animals' eyesight precludes them from red-colour vision? I'm thinking about how night-time field work often involves red torches to minimise disturbance to wildlife. Towards the end of the discussion, this very idea is floated in another context but not in the context of the results of this study.

The second concern is that the text is missing the duration of the experiments. This is critical. It's hard to judge if this dark-seeking behaviour is simply a stress response from being handled, or if over time (e.g. days or weeks, like they would in aquaculture) become accustomed and move to another colour. It is impossible to judge if the results here are a good recommendation for aquaculture to enact without knowing how long the trials were completed for. I also think it would be remiss to not discuss the fact that future studies should examine the growth rates and survival metrics of lobsters reared in these different colours - there are often differences in long term exposure compared to acute effects.

Without addressing these two points, I'm hesitant to state that the results should inform aquaculture practice. Once these have been dealt with, it should be a good paper with clear results.

Specific comments:

- * Ln 33 - Delete "lobster-1".
- * Ln 36 - the methods say five metrics were measured.
- * Ln 39-40 - across these two lines, three different terms are used to describe the same experiment: "follow-on trial", "sub-trial 2" and "follow-up trial". I suggest selecting one and using it throughout the manuscript to help the reader follow along.
- * Ln 40-41 - later in the results/conclusions, the text states red colour was preferred. But as I describe later, I'm not convinced by this result. Either way, the abstract should reflect the same as the main text.
- * Ln 48-49 - I think this opening sentence should have a source for this number.
- * Ln 54 - I think at this stage 'colour' needs more explanation - perhaps "enclosure colour".
- * Ln 74-76 - I don't understand this sentence at all.
- * Ln 80-89 - the first half of this paragraph feels out of place. If you were to follow the usual broad-to-narrow introduction format, these lines could appear immediately after the first paragraph. I.e.

start by discussing colour in general, and then become more specific to talk about the importance of the colour of the background.

- * Ln 115-120 - what were the dimensions of these enclosures?
- * Ln 119 - how were these colour sections arranged? Randomly, in a specific order each time, or randomised design to ensure all possible combinations of colour order were achieved?
- * Ln 130-137 - again need to know the dimensions of the coloured enclosures. Was the 15cm water mark above the top of the colour sections, so water flow was achieved, or were the colour segments porous so water could flow through them? (The latter should be explained in section from Ln 115-120).
- * Ln 130-132 - it might be worth commenting on this filtering/sterilisation step in terms of removing algae (and colour) from the water. I.e. why was this done here?
- * Ln 140-142 - this opening sentence is a bit of a mess...please clean it up. Also at this stage you haven't explained trial number 2 so it doesn't make sense to give the lobster stats on it yet.
- * Ln 147 - missing the word "in" after "observed".
- * Ln 149 - Is the 'behavioural response test' sub-trial 2? Please be specific.
- * Ln 161 - what was the criteria for when the trial ended? This is a critical piece of missing information.
- * Ln 178 - 188 - was each lobster used in multiple trials or just once? Can you explain which data were paired?
- * Ln 201 - remove "an" before "innate".
- * Ln 200 - 202 - It's not clear to me if this "central question" arises from the results of the present study or a setup (or 'gap') for the study?
- * Ln 215 - requires a reference.
- * Ln 221 - This instance of Nephrops is not italicised.
- * Ln 191 - 226 - this whole (big) paragraph feels out of place. At first it seems it should be in the introduction as it is providing background information, but it seems to slowly morph into discussing the results of the study, which have not been presented yet.
- * Ln 234 - 236 - are these differences significant? What are the results of the statistical tests here?
- * Ln 237 - 238 - it's not clear what the link between the results and this pigment is.
- * Ln 242 - 243 - because of camouflage?
- * Ln 243 - 247 - this whole claim seems spurious at best. Are these lobsters likely to have the same neurological pathways that get stimulated by blue light like in humans? What are "healing and calming" properties? How does red light mimic restorative cues?
- * Ln 254-256 - by this stage in this paragraph I am well and truly lost. I think this is a completely new topic to what was discussed above - if so, a new paragraph should be started.
- * Ln 257 - how does red colour simulate natural refuge? Earlier the text says red is unlikely to be experienced underwater.
- * Ln 258 - should be "increased" not "increase".
- * Ln 264 - what are "these findings"?
- * Ln 266 - is this talking about the previous references? It's not clear.
- * Ln 276-277 - but the only darker colour than blue (according to Fig 4) is black. Blue is darker than red, so why would red represent refuge because it's 'dark'?
- * Ln 280-288 - why are these references discussed here and not in the earlier paragraphs?
- * Ln 287 - should "aquaculture system design" be "an aquaculture system designed"?
- * Ln 284-286 - why is this notable? What did the shy fish do?
- * Ln 305 - missing "were".
- * Ln 309 - 310 - again, but blue is more 'black' than red according to figure 4. Also the results of figure 4 really should have been presented before the preference for darkness/blackness is discussed.
- * Ln 311 - 312 - again, this fixed cut off point is not provided but is a critical piece of information.
- * Ln 312-313 - is this simply saying that red and black were preferred by the lobsters? This has been stated many times already.
- * Ln 314 - 315 - this is technically correct but the effect size is so small that I question its meaning. Especially because all other statistics for red vs black are insignificant, including the data contained in 3A which is the exact same data (because there are only two colours) just presented slightly differently.
- * Ln 316-319 - this text belongs in the methods.

- * Ln 320 - 322 - I think there is some confusion around R2 here. Why was linear regression not performed here? I suspect there is a significant relationship on the slope for all 3 metrics and L*.
- * Ln 324 - Can you add error bands to these plots? Also, generally, the response variable is put on the y-axis by convention.
- * Ln 328 - the end of this sentence is repeated. Please proofread. This error is repeated in lines 335 and 342.
- * Ln 345 - delete "Despite these results"
- * Ln 346 - 347 - this has been mentioned a few times so I think it should be clarified - is there a difference between red lighting and red substrate? In this instance I imagine the red coloured segment doesn't truly provide red lighting because the other colours will be reflected into the segment? So I suspect experiments adjusting the lighting colour directly are similar but not directly equivalent.
- * Ln 353-355 - the text has already discussed darkness and refugia given the results. The results should be combined and discussed as a coherent story, rather than provide the same conclusions in multiple places.
- * Ln 376 - 382 - it's not clear why this string of references and descriptions of previous studies is listed. It presents as a list of cherry-picked studies - why is it here? The text has established multiple times that colour preferences have been observed in other species.
- * Ln 399 - what is "detection"?
- * Ln 389 - 426 - there are far too many ideas in this paragraph and it flips multiple times between specific studies, recommendations for future studies, and recommendations for aquaculture. Please just pick one idea per paragraph, clearly explain the evidence for it, and offer your recommendation.
- * Ln 419 - 422 - this is the first thing I thought of when I saw the results so am surprised it's taken this long to touch on it, and the opportunity to explain the results seems to have gone missing. "Red light is visually insensitive to lobsters" - surely this is important for explaining your results?
- * Ln 422 - 426 - how are these random facts related to the previous points?
- * Ln 427 - 444 - there is too much of this "literature review" content in the discussion. It's very verbose now and these points need to be clearly and concisely linked to your results.
- * Ln 445 - delete "significant".
- * Ln 448 - 449 - again, why not blue then?
- * Ln 457-458 - where is the evidence for this statement?
- * Ln 468 - 469 - I'm not convinced that red was preferred over black. This also goes against your conclusion of seeking out 'darker' refuges.
- * Ln 472 - this study did not test survival or growth. Delete this.

Reviewer 2

Comments for the author

The authors investigated background color preferences in a group of juvenile European lobsters. They reported that the lobsters displayed a clear, and similar, preference for black or red backgrounds, compared to blue, green, white and yellow, with potential implications for aquaculture design and animal welfare.

Overall, I found this to be an interesting and useful study, although at times, I did find it difficult to follow the authors train of thought and to contextualize key points. I have made a number of suggestions below that I hope will be useful to the authors.

* I suggest that the authors include a focused results section. Although there are certainly situations where the integration of results can discussion can work well, in this case, I found it difficult to form a clear impression of the overall study findings, and there were certain parts of the discussion that I found difficult to follow, mainly because I was not yet privy to the results on which they were based.

* Overall, I found the manuscript over-long and to contain a lot of apparently extraneous information. It is clear that the authors have extensive knowledge of their subject, but the abundance of information, much of which was not explicitly linked to the study itself, meant that I

sometimes found myself getting lost in the detail. I believe that the manuscript would be stronger and more impactful if it were more succinct and clearly focused on the study's objectives. For example, rather than presenting a full overview of color preference and their biological implications across multiple species, I believe it would be more effective to focus on highlighting only key information necessary to understand the study rationale, the results, and their implications. Where research from other species, or concepts or outcomes that were not directly investigated, are included, their relevance to the current study should be explicitly stated. I believe that this approach will help readers to better understand the study rationale and to contextualize what they are reading.

* Line 56 - 57: This sentence appears to introduce the topic of the next paragraph (i.e., how animals respond to environmental background colors), and it may be preferable to insert the paragraph break before, rather than after it.

* Line 70 - 71: From the current phrasing, I assumed that the cited study evaluated lobster physiology and behavior in response to the color of the cultivation tank, but it seems that the study actually investigated thinlip mullet larvae. I suggest that the authors clarify this point in the text, as similar responses between such different species cannot be assumed.

* Line 74 - 77: I found this sentence somewhat confusing, and suggest rephrasing it to enhance clarity. What did you mean when you said that the study "used pollack as a visual model for monitoring carapace colourimetry concerning camouflage"? I believe the pollack may have been used as model predators to assess how effectively the lobsters blended into different backgrounds? As currently written, the next sentence appears disconnected, as camouflage and photophobic behavior come across as separate concepts. If I have understood well, the intended point is that photophobia represents a mechanism (i.e., light avoidance and preference for darker backgrounds), while enhanced camouflage represents the ultimate adaptive significance of this behavior. Is this correct? If so, some text revision would be useful to clarify the point and to make its relevance to the study more explicit.

* Line 101 - 102: Here, and throughout, I suggest tempering statements made regarding the potential implications of these findings, e.g., for development, physiological plasticity or restocking outcomes. I agree that a better understanding of the lobsters preferences has the potential to impact these outcomes, but this cannot be ascertained based on the current study, and as such, definitive statements on the broader implications of these results cannot be made. Small adjustments, e.g., saying "may" rather than "will", would help to convey better that these are hypotheses rather than established facts. Furthermore, I suggest that it would be useful in the discussion to highlight the importance of further research to evaluate the biological relevance of the color preferences observed. For example, on Line 385 - 388, a follow-up research line regarding how different color tones may influence lobster preference. This may certainly be interesting, however it is important to acknowledge that this study investigated their acute preferences, and I believe that longer-term follow-up studies that investigate the biological implications of these color preferences would be very interesting, and in my opinion, should be a higher priority than investigating the nuance of different color tones.

* Section 2.4. How were the numbers for each trial decided? Am I right in thinking that each of the lobsters took part in a single trial (n = 60 for trial 1 and n = 30 for trial 2)? How long did each trial last?

* Line 161: It would also be useful to comment on the extent of activity, and how it varied across the experimental period. Was the last color chamber occupied at the end of the trial really where the lobsters "settled", or did they tend to continuously move throughout the study period? In the discussion (Line 311 - 312) a comment was made about the video analysis using a fixed cut-off point, which may not fully capture the temporal dynamics of settlement behavior, but it is difficult to contextualize this information without further details on how the experiment was conducted.

* Lines 191 - 196: These opening lines appear more suited to the introduction, rather than the discussion. Indeed there is no explicit mention of the actual study findings in these opening paragraphs of the discussion, which makes it difficult to contextualise this information. As

mentioned above, I suggest that you include a dedicated results section. It would then be useful to use the opening paragraph of the discussion to summarise the key findings from this results section, and then to explore each of these findings in detail throughout the main body of the discussion. Currently, the direct relevance of certain discussion sections to understanding the specific study results are not clear, and it would be useful to be more explicit about how the results obtained informed this discussion.

* Line 201 - 202: From what I understood from section 2.1, the juvenile lobsters used in this experiment were all reared in captivity. As such, I am not clear on how the color preferences observed could represent a learned behavior?

* My understanding was that the main finding was that the juvenile lobsters presented a clear, but equal, preference for red or black backgrounds over the other colors investigated. Yet Line 309 refers to a behavioral inclination toward black backgrounds, whereas line 468 - 469 indicates that red was the most preferred. Some clarity on what the authors consider to be the primary finding from their study would be useful. To reiterate the above point, I believe that a dedicated results section, followed by a clear statement of the main findings that arose from those results in the opening paragraph of the discussion would be very useful.

Reviewer's Responses to Questions

Experimental quality

Does each figure have the proper controls?

If 'No', please indicate reasons in Comments for Author box below.

Reviewer #1:

- Yes

Reviewer #2:

- Yes

Were the data analyzed using appropriate statistical tests?

If 'No', please indicate reasons in Comments for Author box below.

Reviewer #1:

- No

Reviewer #2:

- Yes

Reproducibility

Were experiments performed using adequate number of biological replicates?

If 'No', please indicate reasons in Comments for Author box below.

Reviewer #1:

- Yes

Reviewer #2:

- Yes

Does the methods section provide sufficient detail to permit reproducibility?

If 'No', please indicate reasons in Comments for Author box below.

Reviewer #1:

- No

Reviewer #2:

- No

Completeness

Are the manuscript's conclusions supported by the data?

If 'No', please indicate reasons in Comments for Author box below.

Reviewer #1:

- No

Reviewer #2:

- Yes

Scholarship

Do the authors cite and discuss the merits of data that would argue for and against their conclusion?

If 'No', please indicate reasons in Comments for Author box below.

Reviewer #1:

- No

Reviewer #2:

- Yes

Does the manuscript title & abstract accurately reflect the contents of the manuscript, without hyperbole?

If 'No', please indicate reasons in Comments for Author box below.

Reviewer #1:

- No

Reviewer #2:

- Yes

First revision**Author response to reviewers' comments****Comments from the Reviewers:****Reviewer 1****General comments:**

This manuscript describes useful, clear experiments to test the preference of substrate colour in juvenile European lobster. The experiments themselves appear robust and the results are clear. However, I have major comments on the presentation of the results, their interpretation, and their discussion.

Unfortunately, I don't think the combined results and discussion section works in this instance. Much of the discussion occurs before the reader has been shown the relevant results, and then the same discussion points are re-hashed and repeated later. I find this section far too verbose and repetitive, with many paragraphs that are long and cumbersome. I recommend re-writing the results section clearly and completely, and then moving onto the discussion with clear point-by-point interpretations and recommendations. There is no need to make the same suggestion multiple times. Similarly, there is no need for whole paragraphs that appear to provide a review of the literature without explaining how the mentioned studies relate to the present study.

In terms of interpreting the results, I have two major points of concern: The first of which the text circles around but I don't think ever truly lands on. There is a clear preference for black and red segments, which the authors state is because they represent dark refuges that mimic the "crevice-seeking" behaviour of lobsters. The preference for black segments seems intuitive in that an animal seeks shelter, especially given the stress of handling. However, their own measurements show that the blue segments are darker (or more black) than the red but the text does not offer any explanation for why red is significantly preferred over blue. Is it possible that the animals' eyesight precludes them from red-colour vision? I'm thinking about how night-time field work often involves red torches to minimise disturbance to wildlife. Towards the end of the discussion, this very idea is floated in another context but not in the context of the results of this study.

The second concern is that the text is missing the duration of the experiments. This is critical. It's hard to judge if this dark-seeking behaviour is simply a stress response from being handled, or if over time (e.g. days or weeks, like they would in aquaculture) become accustomed and move to another colour. It is impossible to judge if the results here are a good recommendation for aquaculture to enact without knowing how long the trials were completed for. I also think it would be remiss to not discuss the fact that future studies should examine the growth rates and survival metrics of lobsters reared in these different colours - there are often differences in long term exposure compared to acute effects.

Without addressing these two points, I'm hesitant to state that the results should inform aquaculture practice. Once these have been dealt with, it should be a good paper with clear results.

We thank the reviewer for the constructive comments, which helped us to significantly improve the clarity and structure of the manuscript. We have now separated the results and discussion sections, clarified the interpretation of red versus blue preference, specified the duration of the behavioural trials, and streamlined the discussion to better connect external studies with our findings. Detailed responses to each comment are provided below.

Specific comments:

* Ln 33 - Delete "lobster-1".

Author's response: this has now been removed.

* Ln 36 - the methods say five metrics were measured.

Author's response: We used 3 metrics to assess 5 behavioural parameters.

* Ln 39-40 - across these two lines, three different terms are used to describe the same experiment: "follow-on trial", "sub-trial 2" and "follow-up trial". I suggest selecting one and using it throughout the manuscript to help the reader follow along.

Author's response: sub-trial 2 is now used and has replaced the other two phrases.

* Ln 40-41 - later in the results/conclusions, the text states red colour was preferred. But as I describe later, I'm not convinced by this result. Either way, the abstract should reflect the same as the main text.

Author's response: Now amended: "Sub-trial 2 indicates that the preference for red is marginally greater than black under the same metrics" and thus is in line with the results/conclusion's statements.

* Ln 48-49 - I think this opening sentence should have a source for this number.

Author's response: Now added Hinchcliffe et al., 2022.

* Ln 54 - I think at this stage 'colour' needs more explanation - perhaps "enclosure colour".

Author's response: Now changed to enclosure colour (line 57).

* Ln 74-76 - I don't understand this sentence at all.

Author's response: Now changed to: "The colour of the tank also alters the physical light environment by influencing intensity, reflectance, and absorption (Lesmana et al., 2021). Such changes in lighting can affect lobster welfare, as prolonged exposure to intense illumination may trigger stress responses and increase mortality (Fitch and Lankford, 2013)." (Lines 89-92)

* Ln 80-89 - the first half of this paragraph feels out of place. If you were to follow the usual broad-to-narrow introduction format, these lines could appear immediately after the first paragraph. I.e. start by discussing colour in general, and then become more specific to talk about the importance of the colour of the background.

Author's response: Agreed and now been moved up to a more appropriate spot. - 3rd paragraph.

* Ln 115-120 - what were the dimensions of these enclosures?

Author's response: Now added: "(width: width: 115.4 mm, height: 50 mm, real available volume of one hexagon: 33.98 cm³)". (Lines 129-130). I have also added in another diagram to Figure 1 with the dimensions of 1 part of the hexagonal.

* Ln 119 - how were these colour sections arranged? Randomly, in a specific order each time, or randomised design to ensure all possible combinations of colour order were achieved?

Author's response: Thank you for highlighting this important methodological detail. The colour sections were arranged in a fully randomised order for each trial to avoid positional bias. This ensured that each colour appeared in all possible locations across the experiment, reducing the risk of confounding spatial effects. We have added this clarification to the Methods section. (section 2.2). (Lines 129-132)

* Ln 130-137 - again need to know the dimensions of the coloured enclosures. Was the 15cm water mark above the top of the colour sections, so water flow was achieved, or were the colour segments porous so water could flow through them? (The latter should be explained in section from ln 115-120).

Author's response: Yes, it was above, by 10 cm. Now also made evident in lines 144-146: "The treated seawater was used to fill a shallow tray tank (52 x 42 cm) to a height of 15 cm, 10 cm above the top of the 3D printed colour chambers and was used to host the hexagonal chambers for the colour preference trials"

* Ln 130-132 - it might be worth commenting on this filtering/sterilisation step in terms of removing algae (and colour) from the water. I.e. why was this done here?

Author's response: See lines 141-142.

We use 10 nm particulate filter and ultraviolet light steriliser (35.40 ppt, Seiki Electromechanical Equipment Ltd, Hung Hom, Hong Kong).

* Ln 140-142 - this opening sentence is a bit of a mess...please clean it up. Also at this stage you haven't explained trial number 2 so it doesn't make sense to give the lobster stats on it yet.

Author's response: Now reads: To avoid water disturbance to the lobsters, gentle aeration was supplied into the tanks and was placed away from the hexagonal chamber. Stats are not on the lobsters but on the water temperature and lux and shallow tray tanks.

* Ln 147 - missing the word "in" after "observed".

Author's response: Thank you. Now added.

* Ln 149 - Is the 'behavioural response test' sub-trial 2? Please be specific.

Author's response: Yes. Now reads: The sub-trial 2 behavioural response test was performed 30 times (n = 30).

* Ln 161 - what was the criteria for when the trial ended? This is a critical piece of missing information.

Author's response: This has been revised to 'The durations, incidental movements, and final colour settlement in colour chambers over the 4 hours were recorded for later extrapolation. Five key behavioural metrics were quantified: 1) Mean duration, the average time spent per entry into each chamber; 2) Total duration, the total time spent in each colour chamber per individual; 3) Colour preference ratio, calculated by dividing the mean duration per entry for each colour by the total mean duration across all colours for that individual; 4) Final colour settlement, the last colour chamber occupied at the end of the trial; 5) Incidental movement, the number of times a lobster

entered each colour chamber. These values were calculated for each colour chamber and individual lobster, then compiled into a comprehensive dataset for further analysis.'

* Ln 178 - 188 - was each lobster used in multiple trials or just once? Can you explain which data were paired?

Author's response:

Each animal was used once so in total 90 individuals were used. This is clarified in Ln157, Ln 163 and Ln 168.

* Ln 201 - remove "an" before "innate".

Author's response: Done

* Ln 200 - 202 - It's not clear to me if this "central question" arises from the results of the present study or a setup (or 'gap') for the study?

Author's response: We thank the reviewer for this helpful comment. The intention of this sentence was to highlight a broader research question prompted by the current findings rather than a hypothesis directly tested in this study. To clarify this, we have revised the text to specify that this question emerges as an implication for future research rather than from our own experimental design.

* Ln 215 - requires a reference.

Author's response: The reference Bruno et al., 1977 was added, in accordance.

* Ln 221 - This instance of *Nephrops* is not italicised.

Author's response: Thank you for the observation. It has now been changed.

* Ln 191 - 226 - this whole (big) paragraph feels out of place. At first it seems it should be in the introduction as it is providing background information, but it seems to slowly morph into discussing the results of the study, which have not been presented yet.

Author's response: Thank you for this helpful observation. We agree that the former structure, in which background information and analysis were combined in the same section, made the manuscript difficult to follow. To address this issue, we have now separated the Results and Discussion into two clearly defined sections. This revision ensures that the reader encounters all relevant findings before any interpretation is made in the discussion. Additionally, we restructured and retitled the former results/discussion hybrid paragraphs so that the Results section now focuses solely on empirical findings from the study, including graphical and statistical summaries of settlement preference and movement data, and all literature context on lobster visual ecology and spectral perception now appears under the new Discussion section (4.1). Regarding the paragraph in question, we have converted into a focused subsection (4.1), clearly signposted and used exclusively to interpret the findings.

* Ln 234 - 236 - are these differences significant? What are the results of the statistical tests here?

Author's response: Statistical comparison was not performed for this metric because the values represent pooled durations across all individuals

* Ln 237 - 238 - it's not clear what the link between the results and this pigment is.

Author's response: We thank the reviewer for pointing out the need for clarification regarding the proposed link between red preference and astaxanthin. We agree that the connection was not clearly articulated in the previous version. Our intention was to suggest that the widespread presence of astaxanthin, a red carotenoid pigment found in many crustaceans, may reflect an evolutionary adaptation or internal bias that could influence colour-related behavioural responses. To clarify this, we added the warranty for further investigation to study this potential link.

* Ln 242 - 243 - because of camouflage?

Author's response: We agree that camouflage is a highly relevant factor in substrate selection for benthic crustaceans. In response, we have revised the text to include camouflage as a potential explanation for the observed red preference, noting that red pigmentation and its visual relevance may assist lobsters in concealing themselves within dark or reddish substrates. The revised

sentence now connects evolutionary traits, visual ecology, and behavioural motivations more clearly.

* Ln 243 - 247 - this whole claim seems spurious at best. Are these lobsters likely to have the same neurological pathways that get stimulated by blue light like in humans? What are "healing and calming" properties? How does red light mimic restorative cues?

Author's response: We can agree that the previous version may overstate the relevance of human photobiological studies to crustacean sensory processing. There may not be current evidence supporting shared neural mechanisms underlying blue/red light effects between lobsters and humans. As such, to improve accuracy and avoid unsupported comparisons, we have removed speculative claims about "calming" or "restorative" effects of red light in lobsters. The revised paragraph now focuses on mechanistic explanations grounded in crustacean ecology and visual perception, such as shelter-seeking behaviour, which offer more biologically plausible interpretations of the observed colour preferences.

* Ln 254-256 - by this stage in this paragraph I am well and truly lost. I think this is a completely new topic to what was discussed above - if so, a new paragraph should be started.

Author's response: Thank you for the observation. This paragraph was actually deleted amidst the re-structuring of the section.

* Ln 257 - how does red colour simulate natural refuge? Earlier the text says red is unlikely to be experienced underwater.

Author's response: We agree that the original sentence contained an inconsistency and might have implied a naturalistic role for red colour. In response, we have removed the original sentence. The revised text now clarifies that the preference for red and black backgrounds is more likely associated with low-luminance and high-contrast environments that mimic shelter-like conditions, rather than a direct representation of natural red light underwater.

* Ln 258 - should be "increased" not "increase".

Author's response: Thank you for the attention to that detail. It was corrected.

* Ln 264 - what are "these findings"?

Author's response: In this context, it was intended to refer to the results of the present study on European lobster, which align with previous research on colour-dependent behavioural and welfare responses in other aquatic species. We have now revised the sentence to clarify this connection.

* Ln 266 - is this talking about the previous references? It's not clear.

Author's response:

* Ln 276-277 - but the only darker colour than blue (according to Fig 4) is black. Blue is darker than red, so why would red represent refuge because it's 'dark'?

Author's response: We agree that the original statement may have been implying that red acted as a darker, refuge-like stimulus was inaccurate, particularly in light of the CIELAB analysis, which revealed that blue had a lower luminance than red in our experimental setup. This sentence has now been removed and the revised manuscript instead focuses on the interpretation that red was likely perceived as low-luminance relative to green and white. This also avoids assuming that red inherently resembles natural benthic refuge conditions.

* Ln 280-288 - why are these references discussed here and not in the earlier paragraphs?

Author's response: The additional examples (Nile tilapia, pearl gourami, hybrid catfish and yellow catfish) were discussed in the latter part of the section to extend the discussion from general behavioural comparisons to more applied aquaculture contexts. Earlier paragraphs focused on colour-related behaviour in ecological and evolutionary contexts, while this section specifically addresses practical aquaculture implications, such as welfare, feeding behaviour, and growth responses under different background colours. To clarify this distinction, we have now added a short transitional sentence to make this shift in focus explicit.

* Ln 287 - should "aquaculture system design" be "an aquaculture system designed"?

Author's response: our intention is to state that the findings are important for aquaculture systems design. Thus, in our opinion, the original form is correct.

* Ln 284-286 - why is this notable? What did the shy fish do?

Author's response: we changed the adjective to avoid confusion.

* Ln 305 - missing "were".

Author's response: thank you for the attention to the detail. This was corrected amidst the restructuring of the sections.

* Ln 309 -310 - again, but blue is more 'black' than red according to figure 4. Also the results of figure 4 really should have been presented before the preference for darkness/blackness is discussed.

Author's response: Thank you for highlighting this. Following the suggestion from the reviewer, we revised the manuscript structure to clearly separate the Results and Discussion sections, ensuring that all relevant data, including Fig. 4, are presented before interpretation. We have also clarified in the Discussion (section 4.2) that the preference for red is likely linked to its low luminance under our testing conditions, rather than its spectral hue. This adjustment resolves the earlier ambiguity around "darkness" and improves the coherence between results and interpretation.

* Ln 311 -312 - again, this fixed cut off point is not provided but is a critical piece of information.

Author's response: We have clarified this by amending the sentence to: 'over 4 hours for each lobster experiment' Ln 248-249

* Ln 312-313 - is this simply saying that red and black were preferred by the lobsters? This has been stated many times already.

Author's response: We agree that this sentence was redundant, as the preference for red and black backgrounds had already been clearly communicated through both the Results/Discussion. We have now removed this sentence during the restructuring of the manuscript to improve clarity and avoid unnecessary repetition

* Ln 314 - 315 - this is technically correct but the effect size is so small that I question its meaning. Especially because all other statistics for red vs black are insignificant, including the data contained in 3A which is the exact same data (because there are only two colours) just presented slightly differently.

Author's response: While we agree that the numerical difference between red and black preference ratios is small, the statistical comparison was significant ($P < 0.05$), confirming that juvenile lobsters exhibited a measurable preference for red over black under the conditions tested. We have clarified this in the text without overstating the biological implications. The finding remains relevant for guiding hatchery design, especially where colour-specific tank features are being considered.

* Ln 316-319 - this text belongs in the methods.

Author's response: This was removed during the re-structuring of the section.

* Ln 320 - 322 - I think there is some confusion around R2 here. Why was linear regression not performed here? I suspect there is a significant relationship on the slope for all 3 metrics and L*.

Author's response: The regressions reported in the manuscript were run using linear models, and while R^2 values were presented to highlight the low explanatory power of colourimetric variables overall, the P values for slope significance were examined during analysis and found >0.05 for all behavioural metrics. We did not report these directly in the manuscript to avoid overloading the results section with statistical output, given the absence of significant predictive relationships. However, we are happy to provide the underlying regression summaries if required.

* Ln 324 - Can you add error bands to these plots? Also, generally, the response variable is put on the y-axis by convention.

Author's response: We thank the reviewer for the helpful suggestion. Error bands cannot be added in this case because each point in the scatterplots represents a single mean value per colour

category (n = 1 per colour), meaning there is no variance associated with the plotted CIELAB values. Therefore, error bands are not statistically applicable. Regarding axis orientation, in this case we have retained the current axis order because the CIELAB values function as fixed, non-biological explanatory variables, while the behavioural values represent the dimension of interest. The present layout was chosen to emphasise how colourimetric properties differ across the fixed stimulus set, rather than to imply a fitted statistical mode

* Ln 328 - the end of this sentence is repeated. Please proofread. This error is repeated in lines 335 and 342.

Author's response: We thank the reviewer for the attention to this detail. This was corrected.

* Ln 345 - delete "Despite these results"

Author's response: This was removed during the re-structuring of the section.

* Ln 346 - 347 - this has been mentioned a few times so I think it should be clarified - is there a difference between red lighting and red substrate? In this instance I imagine the red coloured segment doesn't truly provide red lighting because the other colours will be reflected into the segment? So I suspect experiments adjusting the lighting colour directly are similar but not directly equivalent.

Author's response: The reviewer is correct that substrate colour and coloured lighting are not equivalent stimuli. In our study, colour was applied to the substrate only, under constant white lighting. We have added a sentence to clarify this distinction in the manuscript.

* Ln 353-355 - the text has already discussed darkness and refugia given the results. The results should be combined and discussed as a coherent story, rather than provide the same conclusions in multiple places.

Author's response: We acknowledge the reviewer's comment regarding repeated references to darkness and refuge-seeking behaviour. During re-structuring, we removed duplicated statements and streamlined the narrative so that these interpretations now appear only once in the Discussion section, after the relevant results are presented. The Results section now reports the statistical findings only, while the interpretation of results is discussed in a different section.

* Ln 376 - 382 - it's not clear why this string of references and descriptions of previous studies is listed. It presents as a list of cherry-picked studies - why is it here? The text has established multiple times that colour preferences have been observed in other species.

Author's response: We agree that the previously included list of species-specific studies was overly detailed and may not be essential for the main argument. In the revised manuscript, this paragraph has been removed, and the key points now appear more succinctly in new section 4.1, where it is more directly linked to the context of European lobsters. This avoids redundancy and improves the overall focus of the Discussion, in our opinion.

*Ln 399 - what is "detection"?

Author's response: This was removed during the re-structuring of the sections.

* Ln 389 - 426 - there are far too many ideas in this paragraph and it flips multiple times between specific studies, recommendations for future studies, and recommendations for aquaculture. Please just pick one idea per paragraph, clearly explain the evidence for it, and offer your recommendation.

Author's response: Thank you for pointing this out. We agreed that the original paragraph was trying to cover too much at once and became difficult to follow. In the revised version of the manuscript, we've broken this content into separate, topic-focused paragraphs within Section 4.2, and removed some details that were no longer necessary. This has helped to streamline the narrative and keep each paragraph focused on a single idea, improving clarity and flow throughout the discussion section.

* Ln 419 - 422 - this is the first thing I thought of when I saw the results so am surprised it's taken this long to touch on it, and the opportunity to explain the results seems to have gone missing. "Red light is visually insensitive to lobsters" - surely this is important for explaining your results?

Author's response: We appreciate this helpful observation. We agree that the role of red light as a visually insensitive stimulus for lobsters is central to interpreting the behavioural results. In the revised manuscript, we moved this point earlier in the discussion (section 4.1), where we explain the apparent paradox of red being preferred despite reduced visual sensitivity. This repositioning strengthens the logical flow and ensures that the relevance of red light to lobster sensory biology is addressed at the moment it helps clarify the findings.

* Ln 422 - 426 - how are these random facts related to the previous points?

Author's response: We acknowledge the reviewer's concern regarding the transition in this section. The intention of this paragraph was to connect colour-based environmental design with welfare considerations in aquaculture, including how stressors affect lobster health. To improve cohesion, we have revised the paragraph to make this connection clearer by explicitly stating how colour manipulation may mitigate certain stressors associated with high-density systems. No changes were made to the core content, but the contextual link has been strengthened to clarify its relevance to the manuscript overall focus.

* Ln 427 - 444 - there is too much of this "literature review" content in the discussion. It's very verbose now and these points need to be clearly and concisely linked to your results.

Author's response: We appreciate the reviewer's feedback. The intention of this section was to situate our findings within the broader context of colour-mediated physiological and behavioural responses across aquatic taxa. However, we recognise that the literature review structure may appear overly detailed. In light of this, we have revised the paragraph to streamline its content and ensure that each cited study is explicitly linked back to our findings on European lobsters, with a focus on physiological mechanisms and potential applications in aquaculture.

* Ln 445 - delete "significant".

Author's response: This was removed during the re-structuring of the sections.

* Ln 448 - 449 - again, why not blue then?

Author's response: Blue was indeed darker than several other colours tested and fell between red and black in terms of L^* value. However, blue was still consistently avoided by the lobsters, suggesting that low luminance alone does not guarantee preference. Therefore, we have clarified in the manuscript that colour preference in this species likely reflects a combination of luminance and other perceptual or ecological cues.

* Ln 457-458 - where is the evidence for this statement?

Author's response: The statement is based on observed behavioural patterns in our study (preference for dark (black) and red low luminance environments), that resemble natural crevice habitats known to offer refuge from predators (Spanier and Zimmer-Faust, 1988). This behavioural link is also supported by field studies demonstrating the effectiveness of red light for unobtrusive observation due to low visual sensitivity in lobsters (Weiss et al., 2006). We have now clarified this link in the revised manuscript.

* Ln 468 - 469 - I'm not convinced that red was preferred over black. This also goes against your conclusion of seeking out 'darker' refuges.

Author's response: Our statement that red was most preferred refers specifically to the colour preference ratio metric in sub-trial 2, where red showed a statistically significant difference compared to black. Although the absolute difference between red and black was small, the statistical test indicates that the preference for red is not likely due to chance under the conditions tested. At the same time, we acknowledge that both red and black shared similar low-luminance properties and were the most preferred environments overall, supporting the broader interpretation that darker substrates function as surrogate refuges. The statistical outcome does not contradict this but suggests that red may have carried a slight additional cue which we have already discussed in the manuscript.

* Ln 472 - this study did not test survival or growth. Delete this.

Author's response: This was removed during the re-structuring of the sections.

Reviewer 2

The authors investigated background color preferences in a group of juvenile European lobsters. They reported that the lobsters displayed a clear, and similar, preference for black or red backgrounds, compared to blue, green, white and yellow, with potential implications for aquaculture design and animal welfare. Overall, I found this to be an interesting and useful study, although at times, I did find it difficult to follow the authors train of thought and to contextualize key points. I have made a number of suggestions below that I hope will be useful to the authors.

We thank the reviewer for their constructive feedback and positive assessment of our study. We have carefully addressed the specific comments raised below and made revisions to improve the clarity and structure of the manuscript.

* I suggest that the authors include a focused results section. Although there are certainly situations where the integration of results can discussion can work well, in this case, I found it difficult to form a clear impression of the overall study findings, and there were certain parts of the discussion that I found difficult to follow, mainly because I was not yet privy to the results on which they were based.

Author's response: We appreciate the reviewer's recommendation regarding the manuscript structure. In response, we have revised the manuscript to include a dedicated results section separate from the discussion, ensuring that all experimental findings are clearly presented prior to their interpretation. This restructuring improves the logical flow and allows the reader to fully understand the study outcomes before engaging with the broader ecological and aquaculture-related context provided in the discussion. The revised results section now summarises all behavioural metrics, statistical outputs, and CIELAB analyses in a concise format, followed by a streamlined and more focused discussion.

* Overall, I found the manuscript over-long and to contain a lot of apparently extraneous information. It is clear that the authors have extensive knowledge of their subject, but the abundance of information, much of which was not explicitly linked to the study itself, meant that I sometimes found myself getting lost in the detail. I believe that the manuscript would be stronger and more impactful if it were more succinct and clearly focused on the study's objectives. For example, rather than presenting a full overview of color preference and their biological implications across multiple species, I believe it would be more effective to focus on highlighting only key information necessary to understand the study rationale, the results, and their implications. Where research from other species, or concepts or outcomes that were not directly investigated, are included, their relevance to the current study should be explicitly stated. I believe that this approach will help readers to better understand the study rationale and to contextualize what they are reading.

Author's response: We appreciate the reviewer's observation regarding the length and focus of the manuscript. In response, we have carefully revised the introduction and discussion sections to remove extraneous background content and to ensure that the studies cited are aligned with the interpretation of our findings. We have condensed comparative examples and reframed background studies so they more directly support the logical flow and purpose of our work.

* Line 56 - 57: This sentence appears to introduce the topic of the next paragraph (i.e., how animals respond to environmental background colors), and it may be preferable to insert the paragraph break before, rather than after it.

Author's response: Thank you for the observation. It was changed in accordance.

* Line 70 - 71: From the current phrasing, I assumed that the cited study evaluated lobster physiology and behavior in response to the color of the cultivation tank, but it seems that the study actually investigated thinlip mullet larvae. I suggest that the authors clarify this point in the text, as similar responses between such different species cannot be assumed.

Author's response: Thank you for this observation. We agree that the original phrasing could be interpreted as suggesting that the study involved lobsters, when in fact it was conducted on thinlip mullet larvae. We have revised the sentence to clarify the species used in the study cited.

* Line 74 - 77: I found this sentence somewhat confusing, and suggest rephrasing it to enhance clarity. What did you mean when you said that the study "used pollack as a visual model for monitoring carapace colourimetry concerning camouflage"? I believe the pollack may have been used as model predators to assess how effectively the lobsters blended into different backgrounds?

As currently written, the next sentence appears disconnected, as camouflage and photophobic behavior come across as separate concepts. If I have understood well, the intended point is that photophobia represents a mechanism (i.e., light avoidance and preference for darker backgrounds), while enhanced camouflage represents the ultimate adaptive significance of this behavior. Is this correct? If so, some text revision would be useful to clarify the point and to make its relevance to the study more explicit.

Author's response: Thank you for the comment. The reviewer is correct in interpretation. We have revised the sentence in the manuscript to explicitly explain the role of pollack as a predator visual model and to better connect photophobic behaviour with its adaptive significance in terms of camouflage.

* Line 101 - 102: Here, and throughout, I suggest tempering statements made regarding the potential implications of these findings, e.g., for development, physiological plasticity or restocking outcomes. I agree that a better understanding of the lobsters preferences has the potential to impact these outcomes, but this cannot be ascertained based on the current study, and as such, definitive statements on the broader implications of these results cannot be made. Small adjustments, e.g., saying "may" rather than "will", would help to convey better that these are hypotheses rather than established facts. Furthermore, I suggest that it would be useful in the discussion to highlight the importance of further research to evaluate the biological relevance of the color preferences observed. For example, on Line 385 - 388, a follow-up research line regarding how different color tones may influence lobster preference. This may certainly be interesting, however it is important to acknowledge that this study investigated their acute preferences, and I believe that longer-term follow-up studies that investigate the biological implications of these color preferences would be very interesting, and in my opinion, should be a higher priority than investigating the nuance of different color tones.

Author's response: thank you for the suggestions, we have tempered the language to be more hypothetical.

Ln 108-112' Identifying preferred colours may potentially inform the design of lobster aquaculture systems by enabling environmental enrichment that mimics natural habitats, while aiming to maintain appropriate animal husbandry practices, e.g., allowing the hatchery operator to see the animals still. This approach may also help promote physiological plasticity and support juvenile lobsters' acclimation prior to release into the wild, potentially contributing to improved restocking outcomes.'

Given that we split the results and discussion into two subsections. The suggestion is no longer relevant in the section; instead, we have added a suitable sentence to reflect this input.

Ln 406-410.

'addition, longer-term studies (e.g., larvae to pre-release juveniles) should prioritise assessing whether such preferences translate into measurable outcomes in growth, stress physiology, survival, and overall welfare in aquaculture settings. These investigations would help determine whether colour-based enrichment strategies have practical significance beyond short-term behavioural responses.'

* Section 2.4. How were the numbers for each trial decided? Am I right in thinking that each of the lobsters took part in a single trial (n = 60 for trial 1 and n = 30 for trial 2)? How long did each trial last?

Author's response: There have been no prior studies on colour perception in European lobsters, so there was no established guidelines for sample size. To determine an appropriate number, we referred to similar behavioural studies on other crustaceans, including scallop lobsters (Lesmana et al., 2021), where sample sizes typically ranged from 30 to 50 individuals. To ensure sufficient statistical power and account for potential variability in behavioural responses, we increased the number to 60 individuals. This was intended to provide more robust and reliable results in the absence of species-specific recommendations. One lobster was used in each experiment, and there were 60 lobsters/experiments undertaken for subtrial 1 and 30 lobsters/experiments for subtrial 2. We have clarified this in Ln157-159. Each lobster experiment lasted 4 hours and was reported in Ln163.

Lesmana, D. and et al. (2021) The colour preference of scalloped spiny lobster, Panulirus Homarus, Earth and Environmental Science.

* Line 161: It would also be useful to comment on the extent of activity, and how it varied across the experimental period. Was the last color chamber occupied at the end of the trial really where the lobsters "settled", or did they tend to continuously move throughout the study period? In the discussion (Line 311 - 312) a comment was made about the video analysis using a fixed cut-off point, which may not fully capture the temporal dynamics of settlement behavior, but it is difficult to contextualize this information without further details on how the experiment was conducted.

Author's response:

The reference to the last colour choice is stated in Line 186, which clarifies this with '4) Final colour settlement, the last colour chamber occupied at the end of the trial', while the incident movement shows how often the lobsters move in and out of the colour within that period, which is stated in Ln186. Discussion of the extent of the activity is highlighted in Ln 222

* Lines 191 - 196: These opening lines appear more suited to the introduction, rather than the discussion. Indeed there is no explicit mention of the actual study findings in these opening paragraphs of the discussion, which makes it difficult to contextualise this information. As mentioned above, I suggest that you include a dedicated results section. It would then be useful to use the opening paragraph of the discussion to summarise the key findings from this results section, and then to explore each of these findings in detail throughout the main body of the discussion. Currently, the direct relevance of certain discussion sections to understanding the specific study results are not clear, and it would be useful to be more explicit about how the results obtained informed this discussion.

Author's response: Thank you for this helpful suggestion. We agree that the opening of the previous results/discussion may have been too broad. In accordance, we have revised the structure of the new discussion section so that it now opens with a concise summary of the key findings from the experiment, before moving into the interpretation of the results. The general background material that was originally included in the first paragraph of the discussion has been moved into the introduction section, where it is more appropriate indeed.

* Line 201 - 202: From what I understood from section 2.1, the juvenile lobsters used in this experiment were all reared in captivity. As such, I am not clear on how the color preferences observed could represent a learned behavior?

Author's response: The reviewer is correct in noting that the lobsters used in our study were reared entirely in captivity. Our reference to potential learned behaviour was not intended to imply that the lobsters in this study had prior exposure to coloured substrates. Rather, we were referring more generally to the possibility that colour preference may be shaped via experience or developmental plasticity. To avoid confusion, we have clarified this point in the manuscript by specifying that the preferences observed here are unlikely to be influenced by prior environmental exposure, but future work is needed to determine whether colour preference in European lobsters is innate or can be altered through experience.

* My understanding was that the main finding was that the juvenile lobsters presented a clear, but equal, preference for red or black backgrounds over the other colors investigated. Yet Line 309 refers to a behavioral inclination toward black backgrounds, whereas line 468 - 469 indicates that red was the most preferred. Some clarity on what the authors consider to be the primary finding from their study would be useful. To reiterate the above point, I believe that a dedicated results section, followed by a clear statement of the main findings that arose from those results in the opening paragraph of the discussion would be very useful.

Author's response: We appreciate the reviewer highlighting this . In the revised manuscript, we have clarified that while black and red were both strongly preferred over the other colours, red showed a statistically significant advantage over black in the preference ratio analysis in sub-trial 2. We have emphasised both points in the new discussion section.

Reviewer's Responses to Questions

Experimental quality

Does each figure have the proper controls?

If 'No', please indicate reasons in Comments for Author box below.

Reviewer #1:

- Yes

Reviewer #2:

- Yes

Were the data analyzed using appropriate statistical tests?

If 'No', please indicate reasons in Comments for Author box below.

Reviewer #1:

- No

Reviewer #2:

- Yes

Reproducibility

Were experiments performed using adequate number of biological replicates?

If 'No', please indicate reasons in Comments for Author box below.

Reviewer #1:

- Yes

Reviewer #2:

- Yes

Does the methods section provide sufficient detail to permit reproducibility?

If 'No', please indicate reasons in Comments for Author box below.

Reviewer #1:

- No

Reviewer #2:

- No

Completeness

Are the manuscript's conclusions supported by the data?

If 'No', please indicate reasons in Comments for Author box below.

Reviewer #1:

- No

Reviewer #2:

- Yes

Scholarship

Do the authors cite and discuss the merits of data that would argue for and against their conclusion?

If 'No', please indicate reasons in Comments for Author box below.

Reviewer #1:

- No

Reviewer #2:

- Yes

Does the manuscript title & abstract accurately reflect the contents of the manuscript, without hyperbole?

If 'No', please indicate reasons in Comments for Author box below.

Reviewer #1:

- No

Reviewer #2:

- Yes
-

Second decision letter

MS ID#: bio.062203R1

MS Title: Background colour preference in juvenile European lobster (*Homarus gammarus*)

Authors: Matt Bell, Nicholas Kuvaldin, Erik Romero Frontaura, Felix K.A. Kuebutornye, Paula Domech, Carlos Espírito-Santo, Rodrigo O.A. Ozório, Alex Wan

I have now reached a decision on the above manuscript.

The reviewer reports are shown at the bottom of this email.

As you will see, the reviewers gave favourable reports, but raised some critical points that will require amendments to your manuscript. I hope that you will be able to carry these out, because we would like to be able to accept your paper.

At this stage, we also ask you to ensure your manuscript complies with our formatting guidelines - please see our manuscript preparation guidelines for details. Provided you are able to fully address the referees' comments, we are positive about publication of your paper (we accept over 95% of revision submissions) and therefore hope you won't mind any extra work involved in reformatting your manuscript at this point.

Please upload both a 'clean' version of your Word file, along with a highlighted version clearly showing where you have made changes in the revised manuscript. Please avoid using 'Track changes' in Word files as these are lost in PDF conversion.

I should be grateful if you would also provide a point-by-point response detailing how you have dealt with the points raised by the reviewers in the 'Response to Reviewers' box. Please attend to all of the reviewers' comments. If you do not agree with any of their criticisms or suggestions please explain clearly why this is so.

Reviewer 1

Comments for the author

The authors should be commended for taking on board the reviewer comments, which have been incorporated in substantial form throughout the manuscript. I appreciate them restructuring the manuscript, especially the results and discussion, which are now much easier to follow.

I still have a few comments on the writing that I think need to be addressed to make it sufficiently easy for the reader to follow. The discussion is still overly verbose and provides a lot of literature review that doesn't seem to relate to the study's results. Once this is cleaned up, I think the manuscript is ready for publication.

I found this version much easier to follow - well done.

Reviewer 2: I thank the authors for their detailed response to my suggestions. In my opinion, the manuscript is much improved and the separation of the results and discussion section, as well as the increased focus on the study question and findings all helped me to better understand both the results and the author's interpretation.

While reading the manuscript I spotted a couple of minor points that the authors may wish to correct within the final manuscript.

Congratulations on your interesting study.

Specific Suggestions:

Line 107 - 112: The final line is an almost exact replication of that which starts at the end of line 107, so I suggest removing one or other.

Line 168: Parts of this sentence are inaccurate. I suggest removing the term "repeated" behavioral test, as this term is generally used when the same participant/model takes part in each test, which is not the case here. The term "ensure statistical significance" is incorrect, and I assume what you mean to say is that the numbers were intended to ensure adequate power. However without a power calculation this cannot be inferred. It would probably be easiest to just remove this part of the sentence, and simply state that: "A total of 60 independent behavioral tests were conducted, each using a new lobster individual".

Line 173: I'm not sure what you meant by "expect larger effect sizes" at the end of this line? Considering that the results from trial one showed similar preferences between red and black, it could be reasonable to predict smaller effect sizes when these were the only two options. I suggest simply deleting the end of this line to avoid any mis-interpretation. Or if I have misunderstood something here to clarify what was meant by this statement.

Line 226 - 228: It would be useful to present the data from the final settlement preference outcome, e.g., in addition to reporting the chi square results to report the percentage of each lobsters who settled in each of the 5 available colors at the end of the trial. This information may be particularly useful in the case that someone conducts a similar study in the future and wishes to directly compare their results with yours.

Reviewer's Responses to Questions

Experimental quality

Does each figure have the proper controls?

If 'No', please indicate reasons in Comments for Author box below.

Reviewer #1:

- Yes

Reviewer #2:

- Yes

Were the data analyzed using appropriate statistical tests?

If 'No', please indicate reasons in Comments for Author box below.

Reviewer #1:

- Yes

Reviewer #2:

- Yes
-

Reproducibility

Were experiments performed using adequate number of biological replicates?

If 'No', please indicate reasons in Comments for Author box below.

Reviewer #1:

- Yes

Reviewer #2:

- Yes

Does the methods section provide sufficient detail to permit reproducibility?

If 'No', please indicate reasons in Comments for Author box below.

Reviewer #1:

- Yes

Reviewer #2:

- Yes

Completeness

Are the manuscript's conclusions supported by the data?

If 'No', please indicate reasons in Comments for Author box below.

Reviewer #1:

- Yes

Reviewer #2:

- Yes

Scholarship

Do the authors cite and discuss the merits of data that would argue for and against their conclusion?

If 'No', please indicate reasons in Comments for Author box below.

Reviewer #1:

- Yes

Reviewer #2:

- Yes

Does the manuscript title & abstract accurately reflect the contents of the manuscript, without hyperbole?

If 'No', please indicate reasons in Comments for Author box below.

Reviewer #1:

- Yes

Reviewer #2:

- Yes

Second revisionAuthor response to reviewers' comments

Reviewer 1: The authors should be commended for taking on board the reviewer comments, which have been incorporated in substantial form throughout the manuscript. I appreciate them restructuring the manuscript, especially the results and discussion, which are now much easier to follow.

I still have a few comments on the writing that I think need to be addressed to make it sufficiently easy for the reader to follow. The discussion is still overly verbose and provides a lot of literature review that doesn't seem to relate to the study's results. Once this is cleaned up, I think the manuscript is ready for publication.

I found this version much easier to follow - well done.

Author's response: We have gone through the discussion section and reduced what we considered to be less relevant to the manuscript research. If the reviewer would like to reduce more could they give us more guidance.

Reviewer 2: I thank the authors for their detailed response to my suggestions. In my opinion, the manuscript is much improved and the separation of the results and discussion section, as well as the increased focus on the study question and findings all helped me to better understand both the results and the author's interpretation.

While reading the manuscript I spotted a couple of minor points that the authors may wish to correct within the final manuscript.

Congratulations on your interesting study.

Specific Suggestions:

Line 107 - 112: The final line is an almost exact replication of that which starts at the end of line 107, so I suggest removing one or other.

Author's response: We have removed this repetition.

Line 168: Parts of this sentence are inaccurate. I suggest removing the term "repeated" behavioral test, as this term is generally used when the same participant/model takes part in each test, which is not the case here. The term "ensure statistical significance" is incorrect, and I assume what you mean to say is that the numbers were intended to ensure adequate power. However without a power calculation this cannot be inferred. It would probably be easiest to just remove this part of the sentence, and simply state that: "A total of 60 independent behavioral tests were conducted, each using a new lobster individual".

Author's response: thank you for this feedback, we have reduced the sentence as suggested. 'A total of 60 independent behavioural tests were conducted, each using a new lobster (n = 60). '

Line 173: I'm not sure what you meant by "expect larger effect sizes" at the end of this line? Considering that the results from trial one showed similar preferences between red and black, it could be reasonable to predict smaller effect sizes when these were the only two options. I suggest simply deleting the end of this line to avoid any mis-interpretation. Or if I have misunderstood something here to clarify what was meant by this statement.

Author's response: We have deleted the sentence at the end

Line 226 - 228: It would be useful to present the data from the final settlement preference outcome, e.g., in addition to reporting the chi square results to report the percentage of each lobsters who settled in each of the 5 available colors at the end of the trial. This information may be particularly useful in the case that someone conducts a similar study in the future and wishes to directly compare their results with yours.

Author's response: This information was included in the Supplementary Materials section, including the new figures S1 and S2 for both sub-trial 1 and sub-trial 2.

Comments to Author :

General comments:

The authors should be commended for taking on board the reviewer comments, which have been incorporated in substantial form throughout the manuscript. I appreciate them restructuring the manuscript, especially the results and discussion, which are now much easier to follow.

I still have a few comments on the writing that I think need to be addressed to make it sufficiently easy for the reader to follow. The discussion is still overly verbose and provides a lot of literature review that doesn't seem to relate to the study's results. Once this is cleaned up, I think the manuscript is ready for publication.

I found this version much easier to follow - well done.

Specific comments:

Ln 64-66 - this isn't an example of contrast between an animal and its background.

Author's response: this has been removed

Ln 69-71 - delete this, I don't see how its relevant.

Author's response: This has been deleted as recommended

Ln 72- 84 - this whole paragraph seems to go 'backwards' in in the introduction. Earlier the text says it's not all about the colour of the animal but rather the difference in colour between the animal and the substrate, but now we are back to talking about animal colour. The last sentence is the only exception I see. I think this could be combined with the next paragraph and lines 72-80 deleted.

Author's response: We have edited this down and merged with the next paragraph as recommended by the reviewer.

Ln 141 - hanging parenthesis.

Author's response: We have corrected the sentence to-'Note: Figure A and B have perforated holes 2 mm at the bottom to allow water exchange between the tray and the chamber'

Ln 306 - this paragraph needs a topic sentence. I don't really know what the main point is - I think it could be split into 2 paragraphs. I thought it was explaining why red has a stronger response than blue even though blue is a darker colour, but then it goes into a comparison of different lobsters. Pick one point per paragraph please.

Author's response: the section has been split and introductory topic sentence was added at the start of the first paragraph. Line 293 to 311

Ln 342 - is this surprising though? Are you saying the lobsters can't see red wavelengths very well? Which then may mean the red sections seem darker than the blue sections and explain your result.

Author's response: We have updated the sentence to clarify the context: 'As such, any comparison of red-blue light effects across taxa must be interpreted with caution, i.e., lobsters have poor sensitivity to red wavelengths and may therefore perceive red backgrounds as darker than blue ones. Ln 339

Ln 350 - this should be a new paragraph here.

Author's response: This has been moved to a new paragraph was added.

Ln 362 - this paragraph ending here is another huge one which dances around the main point but never seems to actually land. I think this whole section should be explaining why red not blue has such a significant effect. Don't get cute with the wording, just be direct and upfront at the start of the section on what it is explaining.

Author's response: We have removed/reworded the initial sentences to the paragraph to be more direct on the discussion point. 'The preference for red backgrounds in European lobsters could be linked to an innate, phylogenetically conserved trait, potentially associated with camouflage or environmental familiarity during early developmental stages. ' Ln334

Ln 376-377 - which studies?

Author's response: We were referring to the studies mentioned next. In accordance, we have moved this sentence forward.

Ln 382 - 389 - I know all this is basically on the same topic but it's not clear how it relates to your specific results. The discussion section should explain or discuss your specific results, not read as a list of all papers that mention a similar topic. Either make a clear connection or just delete all this.

Author's response: We added these studies in fish to conclude that this preference for such colours in European lobster should not be universally conserved across taxa. As such, we still believe in the importance of such references.

Ln 397 - and the fact that blue is darker than red. I really feel this inconvenience needs to be addressed clearly. You don't need to re-do experiments or analysis, but you need to provide a coherent explanation why you think this happened.

Author's response: the following sentences were added to explain the blue colour lower luminences level, Ln373-378

'A related complication is that, under the study's lighting, the blue chamber had the lowest measured luminance yet was not preferred. This discrepancy likely reflects differences between instrumental luminance and lobster-perceived brightness, i.e., lobsters are more sensitive to blue-green wavelengths (~480-520 nm) and have very poor sensitivity to long-wavelength red light. As a result, a blue surface that is objectively darker may be perceived as relatively brighter, whereas red, despite having a higher L* value, may be perceived as the darkest or most concealing option.'

Ln 411 - 423 - what is the difference in this section's review of the literature compared to the previous section? I'm not clear what the points being made are in each case and how they differ.

Author's response: We have reduced this paragraph and merged it with the previous paragraph to give better clarity to the manuscript.

Ln 424 - 432 - this summary paragraph should be merged with the conclusions section. There is repetition in the two paragraphs and this one already reads like a conclusion.

Author's response: We have removed this section and merged it with the rest of the conclusion section.

Third decision letter

MS ID#: bio.062203R2

MS Title: Background colour preference in juvenile European lobster (*Homarus gammarus*)

Authors: Matt Bell, Nicholas Kuvaldin, Erik Romero Frontaura, Felix K.A. Kuebutornye, Paula Domech, Carlos Espirito-Santo, Rodrigo O.A. Ozório, Alex Wan

I have read through your edits this morning, and I am happy to tell you that your manuscript has now been accepted for publication in Biology Open, pending our standard publication integrity checks. It was accepted on 13th March 2026.